# Beef Cattle Genome Project: Advances in Genome Sequencing, Assembly, and Functional Genes Discovery

**DOI:** 10.3390/ijms25137147

**Published:** 2024-06-28

**Authors:** Zhendong Gao, Ying Lu, Yuqing Chong, Mengfei Li, Jieyun Hong, Jiao Wu, Dongwang Wu, Dongmei Xi, Weidong Deng

**Affiliations:** 1Yunnan Provincial Key Laboratory of Animal Nutrition and Feed, Faculty of Animal Science and Technology, Yunnan Agricultural University, Kunming 650201, China; zander_gao@163.com (Z.G.); yinglu_1998@163.com (Y.L.); 2022004@ynau.edu.cn (Y.C.); mfli_2000@163.com (M.L.); hongjieyun@163.com (J.H.); 15229238680@163.com (J.W.); danwey@163.com (D.W.); dmxiynau@163.com (D.X.); 2State Key Laboratory for Conservation and Utilization of Bio-Resource in Yunnan, Kunming 650201, China

**Keywords:** beef cattle, genome assembly, pan-genome, genes discovery, genetic breeding

## Abstract

Beef is a major global source of protein, playing an essential role in the human diet. The worldwide production and consumption of beef continue to rise, reflecting a significant trend. However, despite the critical importance of beef cattle resources in agriculture, the diversity of cattle breeds faces severe challenges, with many breeds at risk of extinction. The initiation of the Beef Cattle Genome Project is crucial. By constructing a high-precision functional annotation map of their genome, it becomes possible to analyze the genetic mechanisms underlying important traits in beef cattle, laying a solid foundation for breeding more efficient and productive cattle breeds. This review details advances in genome sequencing and assembly technologies, iterative upgrades of the beef cattle reference genome, and its application in pan-genome research. Additionally, it summarizes relevant studies on the discovery of functional genes associated with key traits in beef cattle, such as growth, meat quality, reproduction, polled traits, disease resistance, and environmental adaptability. Finally, the review explores the potential of telomere-to-telomere (T2T) genome assembly, structural variations (SVs), and multi-omics techniques in future beef cattle genetic breeding. These advancements collectively offer promising avenues for enhancing beef cattle breeding and improving genetic traits.

## 1. Introduction

Beef, as one of the primary sources of protein worldwide, plays an irreplaceable role in human diets, whether as a main food item or as a source of various by-products [1]. According to the United States Department of Agriculture (USDA), from 2020 to 2023, global beef production and consumption have continuously increased. This growth is driven by the rising global population and economic levels, which boost the demand for beef, making cattle resources crucial in global agriculture. Despite significant advancements in the beef industry, the genetic resources of beef cattle (*Bos taurus*) face substantial challenges. According to the latest data from the Food and Agriculture Organization of the United Nations [2], there are 1049 native cattle breeds globally, including 110 international transboundary breeds. However, cattle have the highest number of extinct breeds in the report, totaling 159. Therefore, the protection and utilization of beef cattle genetic resources have become critically important.

With the development of high-throughput sequencing technologies, genome sequencing has evolved from the initial Sanger sequencing to second-generation sequencing technologies represented by Illumina, San Diego, CA, USA, and subsequently to third-generation sequencing technologies represented by PacBio, Menlo Park, CA, USA and Nanopore, Oxford, UK. By integrating advanced sequencing technologies and algorithms, numerous fragmented DNA sequences can be accurately assembled into a high-quality reference genome [3]. In March 2022, the telomere-to-telomere (T2T) consortium published the first complete human genome sequence [4], marking another significant milestone following the completion of the human genome draft in 2000 and the fine-scale human genome in 2003. This provides a powerful basis for creating a complete genome map of beef cattle. Therefore, launching the Beef Cattle Genome Project to construct a precise and sensitive functional annotation map of the genome and comprehensively analyze the genetic mechanisms behind critical traits in beef cattle will provide robust genomic data to support precise molecular breeding.

This paper details the advances in genome sequencing and assembly technologies, iterative upgrades of the beef cattle reference genome, and progress in pan-genome research. Subsequently, it discusses studies on the discovery of functional genes associated with key traits in beef cattle, including growth, meat quality, reproductive traits, polled traits, disease resistance, and environmental adaptability. Finally, the prospects of T2T genome assembly, structural variations (SVs) retrieval, and multi-omics combined analysis are explored, providing new perspectives and developmental directions for beef cattle genomics research. This helps deepen the understanding of the structure and function of the beef cattle genome, offering important theoretical support for beef cattle breeding and genetic improvement.

## 2. Sequencing Technologies

First-generation sequencing, known as Sanger sequencing or the “chain termination method,” was pioneered in 1975. Developed by Frederick Sanger, this technique utilized a gel-based method incorporating DNA polymerase, standard nucleotides, and chain-terminating nucleotides (ddNTPs) [5]. During PCR, the incorporation of fluorescently labeled ddNTPs caused random termination, resulting in fragments of varying lengths that ended with A, T, C, or G. Detection of fluorescent signals enabled the determination of the DNA sequence. This revolutionary technology allowed for sequencing fragments of 500–1000 bp and was later automated, replacing large gels with thinner acrylic capillaries. Despite its high accuracy and relatively long read lengths, Sanger sequencing is expensive and inefficient for DNA samples, limiting its scalability [6].

In the early 21st century, next-generation DNA sequencing (NGS) revolutionized genomics by significantly reducing costs and increasing throughput. Led by Illumina, NGS employs short-read sequencing through PCR library construction and laser probe fluorescence signal detection. This method provides high throughput and accuracy, enabling the simultaneous sequencing of hundreds of thousands to millions of DNA molecules. The technology has achieved decreased costs per Gb of data and increased fragment lengths from 30 bp to 300 bp, while maintaining high base accuracy (>99%) [7]. However, NGS has some limitations. PCR amplification and fragment length constraints during library preparation often lead to assembly gaps, with contigs typically only a few tens of kb long. This makes it challenging to resolve complex genome regions rich in tandem repeats, such as centromeres and telomeres, preventing the attainment of a complete genome map [8].

The third-generation sequencing technologies represented by PacBio and Nanopore do not require PCR during library preparation, enabling read lengths of tens of kb to hundreds of kb, and even producing Mb-level reads [9]. With the continuous development of long-read sequencing technologies, read lengths finally reached a critical point: in 2019, PacBio introduced HiFi sequencing technology, with reads up to 15 kb (now up to 23 kb) and an accuracy of 99.9% [10]; in 2021, Oxford Nanopore Technologies (ONT), Oxford, UK, achieved an N50 read length of 100 kb with an accuracy of >98% [11,12].

## 3. Genome Assembly Technologies

In 2023, with the completion of the last piece of the human genome puzzle—the Y chromosome T2T assembly [13]—humanity finally obtained an accurate and complete sequence of its own haploid genome—T2T-CHM13v2.0, marking a milestone in human scientific history. This sequence corrected multiple errors in GRCh38-Y and added over 30 Mb of pairs into the reference sequence, including 110 additional genes, of which 41 are predicted to encode proteins. The comprehensive application of these high-quality template models and technologies will propel beef cattle genomics into the era of “complete map” assembly [11,14].

### 3.1. Assembly Strategies and Steps

Two primary strategies dominate whole-genome sequencing (WGS): the hierarchical shotgun approach and the whole-genome shotgun approach. The hierarchical method involves constructing genetic and physical maps alongside large-insert clone libraries, yielding high-quality sequences but with considerable complexity. Conversely, the whole-genome shotgun approach fragments the genome into smaller pieces for sequencing and subsequent assembly, leveraging advancements in high-throughput technologies (Figure 1) [15].

The whole-genome shotgun approach entails several key steps such as genome prediction, data generation, quality control, sequence assembly, and validation and quality assessment [16]. Addressing repetitive DNA sequences is critical, as these can create ambiguities during assembly, impacting accuracy and continuity [17]. The overall quality and efficiency of genome assembly are influenced by the chosen sequencing technology, assembly algorithms, and the inherent complexity of the genome. Continuous advancements in these areas have markedly improved the accuracy and completeness of genome assemblies, facilitating the study of a growing number of species.

### 3.2. Third-Generation Sequencing Assembly Software

Third-generation sequencing technologies, notably PacBio and Nanopore, offer substantial advantages in genome assembly by producing long reads without the need for PCR amplification. Software tools include Canu, Flye, Falcon, wtdbg2, miniasm, Smartdenovo, MECAT, NECAT, NextDenovo, hifiasm, and others. Several studies have comprehensively compared the performance of various assembly software tools [18,19,20].

The latest research systematically evaluated 11 assembly tools for HiFi sequencing technology. Standard evaluation tools such as QUAST and Benchmarking Universal Single-Copy Orthologs (BUSCO) were used to obtain multiple evaluation metrics, including continuity, completeness, correctness, runtime, and memory usage, which are used to assess the integrity and accuracy of assembly results with reference genomes. The evaluation results showed that the hifiasm and hifiasm-meta became preferred tools for assembling eukaryotic genomes and metagenomes, respectively. In eukaryotic genome assembly, hifiasm exhibited higher continuity, completeness, and accuracy compared to other methods; HiCanu, Verkko, and LJA followed, but Verkko and LJA had shorter contigs. NextDenovo only performed well for haploid genomes [21].

These studies provide clear guidance on how to use high-accuracy long sequence data to assemble complex genomes with high quality. They not only recommend the most suitable assembly tools for relevant research but also point out possible directions for improving assembly algorithms.

### 3.3. High-Quality Genome Assembly

Genome assembly can be achieved at two primary levels: the chromosomal level and the T2T level. These levels differ in accuracy and completeness. The chromosomal level assemblies consist of long contiguous sequences (contigs) that represent different parts of the chromosome but may have gaps or unresolved repeats. This level provides a basic genome structure but can miss complex regions, limiting precise genome comparisons [22].

For the chromosomal level, the assembly focuses on ordering and orienting scaffolds/contigs to reconstruct chromosomes accurately. This level reveals the genome structure and function, including telomeres, centromeres, and gene clusters. It enhances gene annotation, variant detection, and comparative genomics. Additionally, it supports three-dimensional genome techniques, exploring spatial organization and regulatory mechanisms. To achieve chromosomal-level assembly, various auxiliary strategies are used, such as genetic maps [23], Hi-C technology [24], the Chicago and Linked-Reads technologies [25,26], and optical mapping [27]. Consistency comparison of various independent data types such as BioNano optical maps, BAC clones, and genetic maps is used to validate the accuracy of genome assembly from multiple perspectives. Generally, higher consistency indicates more accurate genome assembly.

For the T2T level, T2T genome assembly strives for a high-accuracy, gap-free genome, essential for studying repetitive sequences and complex structures like centromeres and telomeres. T2T genomes offer nearly perfect references, uncovering all genes, structural variations, centromeres, and telomeres, and advancing research on genome function and evolution [28,29]. The development process of T2T assembly has been detailed, highlighting the progress made toward achieving complete genomic assembly from end to end. This work also looks ahead to the necessary technological and algorithmic advancements required to address remaining assembly gaps and to successfully assemble non-diploid genomes. These efforts are crucial for advancing genomic research and ensuring more accurate and comprehensive genome assemblies in the future [30]. Four types of sequencing data are required for T2T assembly. PacBio HiFi sequencing, with read lengths of 10–20 kb and ultra-high accuracy, is crucial for traversing repeats. ONT ultra-long reads (>100 kb) complement HiFi reads by resolving remaining repeats [31]. Hi-C data support chromosome-level assembly, and trio sequencing data aid in genome phasing. For assembly algorithms, tools like Verkko [32] and hifiasm [33] integrate HiFi and ONT data for T2T assembly. For diploid genomes, combining HiFi, ONT, and trio data ensures accurate phasing and chromosome assembly. However, there are some challenges and prospects for T2T genome assembly. Current assembly software relies on algorithms from 1995, which struggle with complex genome regions, leading to gaps. Practical challenges include difficulties in phasing without trio data. Despite recent improvements with PacBio HiFi and ONT data, fully automated T2T assembly remains challenging. Future advancements in sequencing technology and algorithms are expected to achieve complete genome assembly without human intervention [30].

Finally, T2T assembly aims for complete chromosome assembly, the filling of all gaps, and the resolving of repeats to restore genome integrity. This approach offers a more comprehensive and accurate genome assembly, crucial for studying complex genomes [4,14]. In summary, the chromosomal-level assembly provides a foundational genome structure; however, T2T assembly offers comprehensive and high-accuracy genomes essential for advanced genomic research and applications.

### 3.4. Assembly Quality Assessment

Achieving perfect T2T assembly for most animal and plant genomes remains a significant challenge, making post-assembly quality assessment crucial. This evaluation focuses on three key aspects: continuity, completeness, and accuracy. Comprehensive and precise metrics are essential to ensure the assembly meets the standards for further analysis [34,35]. At the same time, it is noteworthy that identifying centromeres and telomeres is particularly important for T2T genomes, as these regions are critical for chromosome stability and function [36].

Continuity is critical in genome assembly, as sequences should be long and contiguous, minimizing fragmentation. Metrics such as N50 for contigs and scaffolds provide insights into assembly quality, with higher values indicating better results. The CC ratio, which compares the number of contigs to chromosomes, is another important measure. A lower CC ratio suggests higher continuity, with a T2T genome ideally having a CC ratio of 1 [35].

Completeness and accuracy are also equally vital. Completeness measures how well the assembled sequences cover the genome. The reads mapping rate reflects the proportion of the genome covered by sequencing data, while BUSCO evaluates completeness by comparing the genome to a database of single-copy orthologous genes. Accuracy assesses how closely the assembled sequences match the true genome. Genome coverage indicates the extent to which sequencing data cover the genome, and tools like Merqury assess consistency between the assembly and the sequencing data, with higher consistency indicating greater accuracy [37].

A recently proposed comprehensive quality evaluation system for genome assembly addresses critical dimensions such as continuity, completeness, accuracy, organelle genomes, and heterozygous genomes. This system includes 14 distinct quality indicators, each with specific benchmarks delineating the criteria for an ideal genome assembly. Employing these metrics provides a robust framework for evaluating assembly quality, ensuring that genome assemblies meet the standards necessary for further biological research and applications [21]. A thorough assessment using these metrics ensures that genome assemblies are suitable for further biological research and applications.

### 3.5. Genome Annotation

Genome annotation is a critical step following genome assembly, involving the identification and characterization of functional elements within the genome, such as genes, regulatory sequences, and structural features. Accurate annotation is essential for understanding the biological functions and evolutionary significance of genomic sequences [38]. It facilitates the identification of genes associated with economically important traits in beef cattle, thereby advancing our knowledge of their genetic basis and enhancing breeding strategies.

Classically, three strategies were employed for gene annotation in the assembly: de novo prediction, homology-based prediction, and transcriptome-based prediction [39]. De novo prediction involves predicting gene structure using existing probability models. This approach has lower accuracy in predicting splice sites and untranslated regions (UTRs). Homology-based prediction leverages the conservation of gene proteins among closely related species. By aligning sequences with high-quality annotation information from these species, it is possible to determine exon boundaries and splice sites. Transcriptome-based prediction utilizes RNA-seq data from the species to assist in annotation, providing precise identification of splice sites and exonic regions [40].

## 4. Genome Assembly Achievements in Beef Cattle

### 4.1. First Beef Cattle Reference Genome

In 2009, an international research team comprising over 300 scientists from 25 countries released the first bovine genome sequence using a sample from the Hereford breed, marking the beginning of bovine genome research. The publication of the first reference genome for Hereford marked a significant advancement in the field of cattle genomics. Over 90% of the assembly is now accurately placed on chromosomes, reflecting a high degree of completeness. The estimated genome size is 2.87 Gb, with 95% of the available expressed sequence tags successfully integrated into assembled contigs, underscoring the assembly’s robustness and reliability for further genomic studies [41]. With the continuous development of second- and third-generation sequencing technologies, in 2017, the beef cattle reference genome underwent its fourth upgrade by integrating Illumina and PacBio sequencing technologies. This version (ARS-UCD1.2) provided a higher-quality genome sequence, characterized by improved continuity and more detailed gene annotation [42]. Currently, the latest version of the beef cattle reference genome is ARS-UCD2.0, which integrates additional sequencing data and further gene annotation, offering more accurate, continuous, and detailed genomic information for beef cattle.

The assembly of reference genomes for various beef cattle breeds has significantly enriched the bovine genomic landscape (Table 1). These genomes not only improve the precision of genomic estimated breeding values in selection programs but also facilitate the discovery of key functional genes related to traits such as growth, carcass characteristics, meat quality, and disease resistance.

### 4.2. Iterative Assembly and Application of Beef Cattle Genomes

The assembly and annotation of beef cattle genomes have been instrumental in exploring evolutionary relationships and population dynamics across different cattle species. These efforts aid in understanding genetic variations, leveraging beneficial traits, and enhancing conservation strategies.

Combining Table 1 and Figure 2, it can be observed that there has been a significant evolution in the sequencing technologies of cattle genomes from 2009 to the present. Initial sequencing efforts utilized shotgun and Sanger sequencing technologies, while subsequent advancements adopted more sophisticated techniques such as the combination of PacBio, Menlo Park, CA, USA and Illumina. Concurrently, there has been an increase in the size of cattle genome sequences, along with improvements in Contig N50 and Scaffold N50, indicating the progress in technology has led to more accurate and complete genome assemblies, continuously improving the quality of beef cattle genome assemblies.

Recent research using Hainan cattle and Mongolian cattle as representative breeds of East Asian zebu and East Asian taurine cattle, respectively, has resulted in the assembly of two high-quality chromosome-level genomes. By integrating population-level second- and third-generation resequencing data, a substantial number of significant SVs influencing the environmental adaptability of Chinese taurine cattle were identified. Cell biology experiments revealed that a 108 bp exon insertion in the *SPN* gene may enhance the uptake of *Mycobacterium tuberculosis* by macrophages by increasing the number of O-glycosylation sites, potentially explaining the low susceptibility of Hainan cattle to bovine tuberculosis. Additionally, 1466 SVs specific to Chinese zebu cattle were identified, likely originating from introgression from Banteng, involving 549 genes. Among these, a 316 bp intronic deletion in the antiviral immune gene *DDX58* was found to have a higher frequency in southern Chinese zebu cattle, which may be linked to disease resistance [51].

These achievements highlight the iterative progress in beef cattle genomics, unveiling genetic diversity, immune system dynamics, and adaptive traits crucial for breed improvement and environmental resilience.

### 4.3. Pan-Genomic Studies of Beef Cattle

The concept of the pan-genome was initially proposed by Tettelin et al. in bacteria [52], referring to the collective genome information of a species or population, consisting of a core genome shared among individuals and an individual-specific dispensable genome. Pan-genomics enables us to trace the evolutionary history of branches and provides new perspectives on the sources of genomic variation and organismal adaptation. In recent years, with the continuous publication of genome data for beef cattle species, pan-genomic studies have made significant progress in genetic improvement, disease resistance, and the optimization of production traits in cattle.

The genomic landscape of cattle has been significantly enriched through various pan-genome initiatives. Initially, the construction of the first graphical pan-genome using data from 288 cattle, encompassing dairy and dual-purpose breeds, improved sequence alignment and variant genotyping accuracy for short-read sequencing [53]. Subsequent efforts integrated third-generation sequencing data from Angus, Brahman, Highland, Brown Swiss, and yak breeds into a high-quality graphical pan-genome, revealing 70 Mb of non-reference sequences, notably 30 Mb from yaks. Transcriptome analysis within these non-reference sequences identified immune-related genes [54]. Expanding on this, data from 294 African cattle breeds facilitated the construction of another graphical genome, uncovering 116 Mb of non-reference sequences that enhanced read alignment and variant detection rates while mitigating allelic biases [48]. Furthermore, utilizing 12 de novo assembled reference genomes led to iterative assemblies of 36 Mb of non-redundant sequences, with significant contributions from yaks. Comparative analysis with the water buffalo genome identified ancestral origins for 83% of these sequences, highlighting enriched proteins related to drug resistance, toxin secretion, signal transduction, and odor receptors [55]. Recent advancements involved resequencing 898 cattle across 57 breeds, revealing 83 Mb of non-reference sequences influencing 279 genes through exon disruption and addition [56]. Moreover, sequencing three hybrid individuals using HiFi and ONT sequencing demonstrated consistent structurally based pan-genomes across different platforms, assembly methods, and coverage ranges [57]. Finally, a super pan-genome of cattle was constructed using haploid resolutions from eight cattle genomes and one reference genome, employing pggb, cactus, and minigraph software tools, thereby enhancing our understanding of genomic diversity, structural variation, and gene function [58]. This effort identified highly variable VNTR regions on chromosome 12, which differed significantly in copy number among groups.

These studies, by integrating genome data from diverse cattle breeds, have uncovered numerous non-reference sequences with important biological functions, including immunity, drug resistance, toxin secretion, and signal transduction. These discoveries are highly significant for beef cattle breeding, offering foundational data and theoretical support for developing superior varieties with enhanced productivity and resistance. The integration of such comprehensive genomic information paves the way for advancements in breeding strategies and the improvement of cattle breeds globally.

## 5. Functional Genes for Important Traits in Beef Cattle

With the rapid development of high-throughput sequencing technologies, the exploration of functional genes and genetic mechanisms underlying important economic traits in beef cattle has evolved from traditional methods to a combination of various omics technologies. WGS has enhanced the evaluation of breeding value and selection processes in beef cattle. By providing a comprehensive and high-resolution view of the genome, WGS enables the identification of genetic variants that influence economically important traits. Successful identification of functional genes and QTLs associated with beef cattle growth, body composition, meat quality, reproduction, disease resistance, and environmental adaptability has been achieved. Each of these areas plays a pivotal role in shaping breeding strategies and enhancing agricultural productivity. These studies provide a solid foundation for a deeper understanding of the genetic regulatory mechanisms underlying important economic traits in beef cattle and for beef cattle breeding efforts.

### 5.1. Growth and Body Traits

Growth and body traits are paramount in beef cattle production, accounting for 60% to 70% of carcass price factors. Genetic improvement is the fundamental approach to beef cattle industry development and also a powerful means to improve meat quality. These discoveries contribute to a deeper understanding of the genetic structure of body traits and will ultimately help us identify potential genes and variations (Table 2).

### 5.2. Meat Quality Traits

Meat quality traits, as complex economic traits, have numerous measurement indicators, but they are interrelated. They are multifaceted and interrelated, encompassing attributes crucial for consumer satisfaction and economic value (Table 3).

Studies at the protein and metabolic levels have shown that muscle color and oxidative stability are closely related to various nutrients and metabolites, such as acylcarnitines, free amino acids, nucleotides, nucleosides, and glucuronate esters [77,78]. Additionally, research on Jinhua cattle has identified genes (such as *MTND2*, *ND4L*, *COII*, *SLC16A7*, and *HOXC6*) related to meat quality traits through transcriptomic and proteomic strategies, along with changes in enzyme activity and metabolite abundance [79,80].

### 5.3. Reproductive Traits

Reproductive traits significantly impact production efficiency and economic outcomes. Oliver et al. found that the *ASIC2* and *SPACA3* genes on *BTA19* are involved in oxidative stress processes and have an impact on pregnancy [81]. Stegemiller et al., through GWAS, identified follicle counts (*AFC*) distributed on chromosomes 2, 3, and 23, while reproductive tract scores (*RTS*) were distributed on chromosomes 2, 8, 10, and 11. In addition, the *STC1* gene is significantly associated with decreased ovulation number, and the transmembrane protein 260 and pyrimidine/pyrimidine monophosphate kinase 2 are closely related to reproductive performance [82]. Interestingly, genes associated with meat quality also affect cattle reproductive performance. For example, allelic genes of *CAPN1* are associated with an increased postpartum estrus interval in beef cattle, *CAST* polymorphism is associated with cattle fertility and reproductive lifespan, and *DGAT1* polymorphism is also associated with cattle reproductive traits. This suggests that using markers like *CAPN1* to improve meat quality may delay re-breeding in cows [83].

### 5.4. Polled Trait

The polled trait is highly desirable in beef cattle due to its implications for animal welfare and operational efficiency [84,85]. The polled variation in cattle was first identified in the Holstein-Friesian breed, located approximately 200 kb downstream of the Celtic variation, spanning from 1,909,352 to 1,989,480 bp, with an 80,128 bp duplication [86]. Currently, candidate mutations PF and PC for the polled trait have been identified in breeds such as Angus, Holstein, and Simmental. The PF mutation mainly exists in Holstein-Friesian cattle, caused by duplication and three SNPs at 80,128 bp on chromosome 1. The PC mutation is mainly found in Angus and Simmental cattle, resulting from a repeat and replacement of the original 10 bp at 202 bp on chromosome 1 [87]. The United States and Germany have successfully introduced the PC mutation using gene editing technology into Holstein and Holstein-Friesian cattle, resulting in the creation of polled new breeds [88,89].

A Mongolian polled variation has been discovered in Mongolian yaks and Mongolian taurine cattle. Whole-genome sequencing of homozygous and heterozygous polled yaks localized the Mongolian polled yak locus to an 800 kb region on *BTA1*. The first mutation is a 219 bp insertion, downstream 61 bp from the original sequence (starting at 1,976,128 bp), and the second mutation is a 6 bp deletion and 7 bp insertion upstream at 621 bp. Within the 219 bp repeat sequence, there are 11 bp motifs (5′-AAAGAAGCAAA-3′) that are fully conserved in the *Bovidae* family, suggesting potential functional significance [90].

### 5.5. Disease Resistance and Environmental Adaptation

Over the past 10,000 years, beef cattle have undergone several major migrations, resulting in genetic exchanges and mixing among cattle populations and their distribution across continents inhabited by humans worldwide. During these migrations, beef cattle rapidly adapted to various extreme environments, including high and low temperatures, high altitudes, drought, and humidity. These adaptive evolutionary processes have led to significant genomic diversity among different cattle breeds [91]. The summarized genes related to disease resistance and adaptation are as follows (Table 4).

## 6. Conclusions and Perspective

The field of genome research has seen remarkable advancements with the continuous evolution of sequencing technologies. From the initial draft genome (Genome 1.0) to the high-quality genome (Genome 2.0) achieved through PacBio and Hi-C technologies, and further to the near-complete genome (Genome 3.0) assembled with ONT sequencing, the journey culminates in the T2T genome (Genome 4.0) using ONT ultra-long reads and PacBio HiFi technology. T2T genomes have now been successfully constructed for minor species, filling existing genome gaps and enhancing the resolution of genetic variation and functional gene identification. This progress serves as a guiding beacon for future precise breeding in beef cattle.

Future advancements in genome assembly will provide critical resources for constructing species pan-genomes, facilitating the exploration of genetic variations across different beef cattle breeds. The integration of genomics with emerging technologies, such as pan-3D genomics, single-cell sequencing, and gene editing, underscores an inevitable trend in genomics research. SVs have emerged as focal points in species’ adaptive evolution studies, particularly within graph-based pan-genome frameworks. Unlike single-nucleotide polymorphisms (SNPs), SVs often exhibit larger dosage effects, influencing multiple genes or regulatory elements simultaneously, leading to pronounced phenotypic variations. Long-read sequencing technologies enable precise breakpoint identification, providing a more scientific and accurate basis for SV detection in beef cattle genome research.

The regulatory network of cattle is highly complex, necessitating a shift from single-omics approaches to integrated multi-omics technologies encompassing genomics, transcriptomics, proteomics, metabolomics, lipidomics, and phenomics. Multi-omics approaches promise comprehensive insights into genetic breeding traits in beef cattle, marking a pivotal shift in research methodologies. The integration of these technologies with bioinformatics and systems biology will be essential for dissecting complex traits and enhancing breeding strategies. As sequencing technologies and computational algorithms evolve, innovations like T2T genomes, structural variations, and multi-omics analyses are paving the way for new frontiers in beef cattle genomics, propelling the industry into “A NEW ERA”.

## Figures and Tables

**Figure 1 ijms-25-07147-f001:**
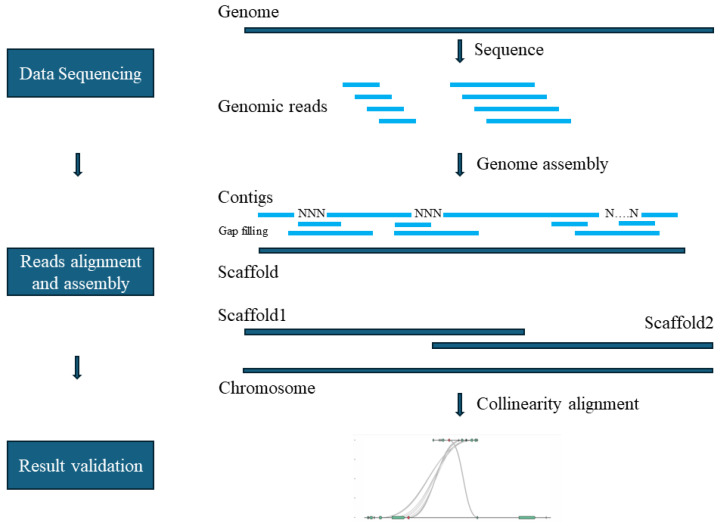
Basic steps of genome assembly.

**Figure 2 ijms-25-07147-f002:**
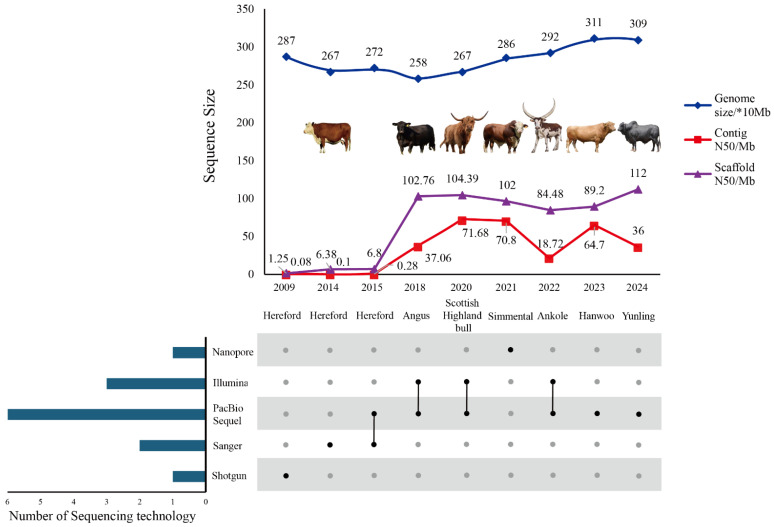
The iteration of cattle genome assembly quality and sequencing technology (gray and black dots indicate whether the method was used. “×10 Mb” represents that the value is multiplied by 10, in units of Mb.). Note: the assembly information of the beef cattle breed with the highest genome quality in each selected year was chosen in the figure, in order to explore the optimal assembly strategy (the beef cattle images in the figure are for illustrative reference only).

**Table 1 ijms-25-07147-t001:** Research progress on genome assembly of beef cattle.

Breed	Genome Version	Sequencing Technologies	Sequence Size (Gb)	Assembling Evaluation Metrics			Year	References
				Contig N50 (Mb)	Scaffold N50 (Mb)	BUSCO		
Hereford	Btau_4.0	Shotgun	2.7	0.76	1.9	--	2009	[41]
	UMD Bos_taurus 2.0	Shotgun	2.87	0.08	1.25	--	2009	[43]
	UMD_3.1.1	Sanger	2.67	0.1	6.38	--	2014	[44]
	Btau_5.0.1	Sanger PacBio RS II	2.72	0.28	6.8	--	2015	GCA_000003205.6
	ARS-UCD1.2	PacBio Illumina	2.72	25.9	103.31	--	2018	[42]
	ARS-UCD2.0	--	2.77	26.4	103.3	--	2023	GCA_002263795.4
Angus	UOA_Angus_1	PacBio Illumina	2.58	37.06	102.76	93%	2018	[45]
Scottish Highland bull	ARS_UNL_Btau-highland_paternal_1.0_alt	PacBio Illumina	2.67	71.68	104.39	97.1%	2020	[46]
Simmental	ARS_Simm1.0	Nanopore PromethION	2.86	70.8	102	90.8%	2021	[47]
Arequipa fighting bull	B.taurus_INIA.1	Illumina	2.81	0.06	110.9	--	2022	GCA_024542955.1
N’Dama	ROSLIN_BTT_NDA1	PacBio Illumina	2.76	11.06	104.85	93.9%	2022	[48]
Ankole	ROSLIN_BTI_ANK1	PacBio Illumina	2.92	18.72	84.48	94.0%	2022	[48]
Hanwoo	SNU_Hanwoo_2.0	PacBio Sequel	3.11	64.7	89.2	95.8%	2023	[49]
Yunling	YAU_Btau_1.0	PacBio Sequel	3.09	36	112	95.8%	2024	[50]

**Table 2 ijms-25-07147-t002:** Overview of the growth and body traits genes in cattle.

Category	Analytical Method	Gene/Variant Locus	Related Function	References
Body conformation	GWAS/ Meta-GWAS	*LCORL*	The linkage disequilibrium between SNPs of the *LCORL*/*NCAPG* genes is relatively high. Additionally, a missense mutation in the *LCORL* gene is significantly associated with cattle body size.	[59]
		*NCAPG*		
		*PENK*	In Chinese Yanbian cattle populations, identified SNPs were found to be associated with traits such as wither height, chest depth, body height, and body length.	[60,61,62]
		*XKR4*		
		*IMPAD1*		
		*CCND2*		
		*SNTG1*		
		*TTC30B*		
		*HMGA1*		
Growth and developmental traits	GWAS/ TWAS	*SOX2*	Associated with traits such as birth weight, daily weight gain, body weight, slaughter rate, and net meat yield.	[62,63,64,65,66,67,68]
		*PLAG1*		
		*RYR*		
		*EML6*		
		*HOMER1*		
		*SAMD12*		
		*IGFBP5*	The expression of muscle *IGFBP5* is significantly associated with daily weight gain traits.	
		*LPIN2*	The expression of liver *LPIN2* is significantly associated with backfat thickness traits.	
		*TIGAR*	Muscle SNPs can regulate the expression of the *TIGAR* gene and are significantly associated with traits such as body height and hind leg circumference.	
		*PRKD3*	They are significantly associated with traits such as body weight and slaughter rate.	
		*FXN*		
Multiple traits	Genome-wide selection-mapping scans	Core selective sweep regions (CSSs)	291 genes were identified within the CSS interval, related to traits such as dairy and meat production, stature, and coat color traits.	[69]

**Table 3 ijms-25-07147-t003:** Overview of the meat quality traits genes in cattle based on GWAS.

Category	Genes	Associated Function	References
Fatty acid/lipid metabolism	*FASN*	Multiple SNPs in genes have significant effects on fatty acids. These genes influence various physiological aspects related to body size, meat quality, and feed intake by controlling muscle regulatory signaling pathways, key signaling molecules like *INS*/*IGF-1* pathways, lipid metabolism, and adipose tissue development.	[70,71,72,73]
	*SCD*		
	*ELOVL5*		
	*CASP2*		
	*CAST*		
	*PLAG1*		
	*PLAGL2*		
	*PLA2G16*		
	*AQP3/AQP7*		
	*MYLK2*		
	*WWOX*		
	*XKR4*		
Associated with meat tenderness, eye muscle area, backfat thickness, marbling score	*CARTPT*	QTL intervals associated with meat color, tenderness, eye muscle area, backfat thickness, shear force, and marbling score have been identified in genes.	[74,75,76]
	*ASAP1*		
	*CAPN1*		
	*CAPN5*		
	*TMEM236*		
	*SORL1*		
	*TRDN*		
	*S100A10*		
	*AP2S1*		

**Table 4 ijms-25-07147-t004:** Overview of known genes under local adaptation in cattle populations.

Category	Breed/Population	Genes	Function	References
Cold adaptation	Yakut cattle	*NRAP*	Amino acid changes in *NRAP* suggest they may slow metabolism but enhance heart function to supply blood during winter.	[92]
	Russian native cattle	*RETREG1*	*RETREG1* is associated with human pain and temperature perception disorders. Positive selection features of *RETREG1* may aid the breed’s adaptation to its harsh environment.	[93]
	Fjall cattle	*AQP3* *AQP7*	Positive selection features have been observed near the *AQP3* and *AQP7* genes in northern Swedish cattle. They encode aquaporins that facilitate transport of water, glycerol, and urea, likely playing a role in temperature regulation via sweating.	[94]
	Mongolian cattle Yanbian cattle	*PRDM16*	A mutation in the *PRDM16* gene increases expression of thermogenic-related genes, maintaining the formation of brown adipocytes, suggesting its crucial role in cold tolerance.	[95]
Drought adaptation	African taurine cattle	*HSPA4* *HSPA9* *DNAJC18* *SOD1*	Heat tolerance is a well-known trait in Indian cattle, with these genes being members of the heat shock protein family involved in crucial cellular defense mechanisms under exposure to hot environments.	[96,97,98]
	Brahman cattle	*FADS2P1*	*FADS2P1* is a pleiotropic gene involved in biosynthesis of unsaturated fatty acids, lipid homeostasis, inflammation response, and promoting muscle cell growth and cell signaling.	[45]
Temperature regulation	Senepol cattle	*PRLR*	Cattle with extremely short and smooth hair exhibit strong heat tolerance, capable of withstanding hot weather, and a mutation in exon 11 of the prolactin receptor (*PRLR*) has been shown to have a significant impact on the smooth hair phenotype in cattle.	[99]
	African taurine cattle	*PRLH*	Prolactin-releasing hormone gene (*PRLH*) shows selective advantage related to heat tolerance in African taurine cattle, with selective advantage in prolactin expression.	[97]
Water resorption	African humped cattle	*GNAS*	The *GNAS* gene is associated with water resorption through mediating the action of antidiuretic hormone arginine vasopressin on aquaporin-2 water channels and subsequently aiding in renal water pathways.	[98]
Local pathogen adaptation	African cattle	*TICAM1* *ARHGAP15* *CARD11*	These genes are involved in signaling in T and B cells of the immune system. *TICAM1* encodes an adapter protein containing the Toll/interleukin-1 receptor (*TIR*) domain, involved in innate immune response against invading pathogens.	[98,100]
	African indicine cattle	*ATG4B* *MATR3* *MZB1* *STING1*	These genes may confer genetic resistance to ticks and tick-borne diseases. Among them, *STING1* regulates the production of type I interferons mediated by intracellular DNA, making it crucial for host defense against DNA pathogens.	[98]
Light-colored coat	Brahman Nelore Gir	*MC1R* *ASIP*	*MC1R* gene in Brahman, Nelore, and Gir cattle shows positive selection. *ASIP* plays a crucial role in reducing eumelanin and increasing pheomelanin production by blocking *MC1R*.	[101,102]
White coat color	Borgou cattle	*SILV*	The *SILV* gene encodes the type I integral membrane protein in premelanosome matrix (*PMEL17*), essential for melanosome development and responsible for diluting or lightening the base color defined by *MC1R* in certain cattle breeds.	[103]
Dark brown coat color	Chinese Yellow cattle	*KITLG* *LEF1* *MCM6*	Involved in UV protection, various coat color, and pigment deposition genes (*KITLG*, *LEF1*, and *MCM6*).	[104]
Black coat color	Zhoushan cattle	*MC1R*	The *MC1R* gene regulates animal melanin synthesis. The dark coat color in Zhoushan cattle may be associated with the p.*F195L* mutation in *MC1R*.	[105]
Hypoxia adaptation	Tibetan cattle	*EPAS1*	*EPAS1* encodes a subunit of the HIF transcription factor, crucial for Tibetan human hypoxia adaptation. Highly divergent nonsynonymous SNPs have been found in *EPAS1* in Tibetan cattle, potentially related to their adaptability.	[106]
Dwarfism	Tibetan cattle	*HMGA2* *ADH7*	*HMGA2* is identified as a candidate gene associated with high-altitude adaptation in humans and livestock. *ADH7* is another gene related to dwarfism in Tibetan cattle, both of which have undergone positive selection.	[106]
Adaptive immune response	Jiaxian Red cattle Bashan cattle Bohai Black cattle	*SLAMF1* *CD84* *SLAMF6*	The regions of *SLAM* family genes, including *SLAMF1*, *CD84*, and *SLAMF6*, may be related to the high disease resistance of indigenous Chinese cattle breeds.	[107,108,109]
Hair growth	Mongolian cattle	*DVL2*	*DVL2* plays a significant role in regulating hair growth and its association with the hair follicle cycle. A significant north–south breed difference in a missense mutation of the *DVL2* gene in Mongolian cattle may affect hair growth and adaptation to different climates.	[110]

## Data Availability

Not applicable.

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
