# Peer review of "Beef Cattle Genome Project: Advances in Genome Sequencing, Assembly, and Functional Genes Discovery"

_ijms, 2024, doi:10.3390/ijms25137147_

Round 1

Reviewer 1 Report

Comments and Suggestions for Authors

In the present work, Gao try to review the beef cattle genome project in genome assembly, functional genes discovery, and genetic breeding. In this manuscript, the latest advances in beef cattle genome assembly, functional gene discovery, and genetic breeding, as well as functional genes and genetic mechanisms related to important economic traits in beef cattle are reviewed. The structure of the article is reasonable, and holds academic value and application prospects. However, there are still some aspects that can be further refined to enhance the readability and impact of the article.

Major concerns

1. English grammar and writing style should be checked and revised throughout the manuscript.

2. As a high impact factor Journal for a review paper, additional fine Figures should be included, which may include the genetic breeding of the beef cattle genome project with image of beef.

3. Assembly Quality Assessment: The detailed introduction of key indicators and tools for post-assembly quality assessment provides a comprehensive evaluation system. This section is valuable, but some repetitive descriptions can be simplified to ensure the focus is highlighted. Regarding Figure 2, ‘Iterative Quality Evaluation of Beef Genome Assembly and Sequencing Technologies,’ ensure its clarity for better reader comprehension. In addition, the recommendation is to delete the sentence ‘In metagenome assembly evaluations, hifiasm-meta and metaflye had the fewest assembly errors, but hifiasm-meta showed superior integrity and continuity compared to metaflye, particularly in the face of complex metagenomes, although it retained some redundant sequences.’

Minor concerns

1. Line 5, deleted ‘and’.

2. Abstract section should be rewritten. Refine the Lines 15-19, and a summary sentence may be added in the end of the Abstract section.

3. There are so many ‘we’ in the Abstract section and Introduction section, which should be revised.

4. Figure 1, for the result validation, the picture may be change to more suitable image.

5. Lines 123-131, two short paragraphs.

6. Line 175, in general, ‘Furthermore,’ is not at the beginning of a paragraph.

7. Lines 184-197, a reference is needed.

8. Lines 210-215, there is no a reference in this paragraph.

9. Lines 304-342, too long paragraph.

10. The format of Tables should be revised.

11. The Conclusions and Perspective section should be refined. There are so many paragraphs and references.

Comments on the Quality of English Language

Extensive editing of English language required.

Author Response

Dear Reviewer:

Thank you for your letter and for the comments concerning our manuscript entitled “Beef Cattle Genome Project: Advances in Genome Assembly, Functional Genes Discovery and Genetic Breeding” (Manuscript ID: ijms-3065515). Those comments are all valuable and very helpful for revising and improving our paper, as well as the important guiding significance to our researches. We have studied comments carefully and have made correction which we hope meet with approval.

We have addressed your major and minor concerns as follows:

Major concerns

The reviewer’s comment: 1. English grammar and writing style should be checked and revised throughout the manuscript.

Response: Thanks for your suggestion. We have tried our best to polish the language in the revised manuscript.

The reviewer’s comment: 2. As a high impact factor Journal for a review paper, additional fine Figures should be included, which may include the genetic breeding of the beef cattle genome project with image of beef.

Response: Thank you for bringing up this issue, we have inserted a high-definition image of the beef cattle in Figure 2.

The reviewer’s comment: 3. Assembly Quality Assessment: The detailed introduction of key indicators and tools for post-assembly quality assessment provides a comprehensive evaluation system. This section is valuable, but some repetitive descriptions can be simplified to ensure the focus is highlighted. Regarding Figure 2, ‘Iterative Quality Evaluation of Beef Genome Assembly and Sequencing Technologies,’ ensure its clarity for better reader comprehension. In addition, the recommendation is to delete the sentence ‘In metagenome assembly evaluations, hifiasm-meta and metaflye had the fewest assembly errors, but hifiasm-meta showed superior integrity and continuity compared to metaflye, particularly in the face of complex metagenomes, although it retained some redundant sequences.’

Response: We have simplified the repetitive descriptions in the Assembly Quality Assessment section to ensure the focus is highlighted. We have also revised Figure 2 to improve its clarity. Additionally, we have removed the sentence ‘In metagenome assembly evaluations, hifiasm-meta and metaflye had the fewest assembly errors, but hifiasm-meta showed superior integrity and continuity compared to metaflye, particularly in the face of complex metagenomes, although it retained some redundant sequences.’

Minor concerns

The reviewer’s comment: 1. Line 5, deleted ‘and’.

Response: We were really sorry for our careless mistakes. Thank you for your reminder and we have deleted 'and' in Line 5.

The reviewer’s comment: 2. Abstract section should be rewritten. Refine the Lines 15-19, and a summary sentence may be added in the end of the Abstract section.

Response: Thank you for pointing this out. The whole Abstract section has been rewritten for better clarity, with refined lines and an added summary sentence at the end.

We have changed Lines 15-29 into: “Beef is a major global source of protein, playing an essential role in the human diet. The worldwide production and consumption of beef continue to rise, reflecting a significant trend. However, despite the critical importance of beef cattle resources in agriculture, the diversity of cattle breeds faces severe challenges, with many breeds at risk of extinction. The initiation of the Beef Cattle Genome Project is crucial. By constructing a high-precision functional annotation map of the genome, it becomes possible to analyze the genetic mechanisms underlying important traits in beef cattle, laying a solid foundation for breeding more efficient and productive cattle breeds. This review details advances in genome sequencing and assembly technologies, iterative upgrades of the beef cattle reference genome, and its application in pangenome research. Additionally, it summarizes relevant studies on the discovery of functional genes associated with key traits in beef cattle, such as growth, meat quality, reproduction, polled traits, disease resistance, and environmental adaptability. Finally, the review explores the potential of telomere-to-telomere (T2T) genome assembly, genome structural variation (SV), and multi-omics techniques in future beef cattle genetic breeding. These advancements collectively offer promising avenues for enhancing beef cattle breeding and improving genetic traits.”

The reviewer’s comment: 3. There are so many ‘we’ in the Abstract section and Introduction section, which should be revised.

Response: The excessive use of 'we' in the Abstract and Introduction sections has been revised, and we polished the language in the Abstract section and Introduction section.

We have changed Line 50 “we can accurately assemble numerous fragmented DNA sequences into a complete beef cattle genome map, enabling the construction of high-quality reference genomes.” into Line 49 “numerous fragmented DNA sequences can be accurately assembled into a complete beef cattle genome map, enabling the construction of high-quality reference genomes.”

Line 61 “In this paper, we detail” into Line 59 “This paper details”.

Line 63 “we discuss” into Line 61 “it discusses”.

The reviewer’s comment: 4. Figure 1, for the result validation, the picture may be change to more suitable image.

Response: We appreciate the reviewer’s comment here, we believe that adding a new figure, as the reviewer suggested, would be unnecessary. It is necessary to use alignment tools to align the reassembled genome in order to evaluate the completeness of assembly and the uniformity of sequencing coverage. The alignment tools should be used to calculate the alignment rate of reads, the degree of coverage of the genome, and the distribution of depth. The results in the figure 1 validate the data alignment process. I hope this answer satisfies you.

The reviewer’s comment: 5. Lines 123-131, two short paragraphs.

Response: Thank you for your valuable feedback. I have combined the two short paragraphs (Please see lines 119-126 of the revised manuscript) into a single, more cohesive paragraph to enhance the readability and flow of the manuscript. I appreciate your attention to detail and suggestions for improvement.

The reviewer’s comment: 6. Line 175, in general, ‘Furthermore,’ is not at the beginning of a paragraph.

Response: Thank you for your insightful comments. I have change Line 175 “Furthermore, for T2T level” into Line 165 “For T2T level”

The reviewer’s comment: 7. Lines 184-197, a reference is needed.

Response: We feel sorry for our carelessness. This section had references when it was written, but they were carelessly deleted during later revisions. The references have been reinserted into the text. Your guidance is greatly appreciated. (Please see lines 177 and 179 of the revised manuscript)

  1. Chaisson, M. J. P.; Huddleston, J.; Dennis, M. Y.; Sudmant, P. H.; Malig, M.; Hormozdiari, F.; Antonacci, F.; Surti, U.; Sandstrom, R.; Boitano, M.; Landolin, J. M.; Stamatoyannopoulos, J. A.; Hunkapiller, M. W.; Korlach, J.; Eichler, E. E. Resolving the complexity of the human genome using single-molecule sequencing. Nature 2015, 517, (7536), 608-611. https://doi.org/10.1038/nature13907
  2. Rautiainen, M.; Nurk, S.; Walenz, B. P.; Logsdon, G. A.; Porubsky, D.; Rhie, A.; Eichler, E. E.; Phillippy, A. M.; Koren, S. Telomere-to-telomere assembly of diploid chromosomes with Verkko. Nat. Biotechnol. 2023, 41, (10), 1474-1482. https://doi.org/10.1038/s41587-023-01662-6
  3. Cheng, H.; Concepcion, G. T.; Feng, X.; Zhang, H.; Li, H. Haplotype-resolved de novo assembly using phased assembly graphs with hifiasm. Nat. Methods 2021, 18, (2), 170-175. https://doi.org/10.1038/s41592-020-01056-5

The reviewer’s comment: 8. Lines 210-215, there is no a reference in this paragraph.

Response: The reference for this paragraph is reference 35, and we have add reference [35] in Line 207.

The reviewer’s comment: 9. Lines 304-342, too long paragraph.

Response: Thank you for your constructive feedback regarding the length of the paragraph in lines 304-342 of our manuscript. We have simplified this paragraph to focus on specific advancements and findings in cattle pan-genomics (Please see lines 309-333 of the revised manuscript)

The reviewer’s comment: 10. The format of Tables should be revised.

Response: Thank you for your comment regarding the format of the tables in our manuscript. The tables have been formatted strictly according to the style guidelines of the International Journal of Molecular Sciences (IJMS). Based on this adherence, we believe the tables are in line with the journal's requirements and do not require further revision.

The reviewer’s comment: 11. The Conclusions and Perspective section should be refined. There are so many paragraphs and references.

Response: We appreciate your feedback and have condensed the content to make it more concise and focused. We have changed Line 425-472 into Line 421-450, and removed the number of references concurrently.

Yours sincerely,

Dr. Weidong Deng

Yunnan Provincial Key Laboratory of Animal Nutrition and Feed, Faculty of Animal Science and Technology, Yunnan Agricultural University, Kunming 650201, China

Tel.: +86-871-65220375

Email: dengwd@ynau.edu.cn

Reviewer 2 Report

Comments and Suggestions for Authors

This paper discusses the progress of Beef Cattle Genome Project: Advances in Genome Assembly, Functional Genes Discovery and Genetic Breeding. My suggestions for this manuscript is: it needs a major revision.

1. Line 105:the title is “Genome Assembly Technologies”, but line 132 is titled “Third-Generation Sequencing Assembly Software”,so is the third part mainly about assembling software or technology?

2. Line184-185:“four types of sequencing data required for T2T assembly” are mentioned in the paper. As we all know, the T2T genome assembly mainly relies on HIC, ONT ultra-long and HIFI reads sequencing data, why did you write about four data types with no references?

3. Line 192-193: please add reference documents to make your argument more convincing.

4. Line204-209: The quality of genome assembly was assessed mainly for continuity, completeness, and accuracy, but for T2T genomes, the identification of centromere and telomere should also be added.

5. Part 3 suggests adding an introduction to the Genome Annotation section.

6. Line240: The full name should be written when EST sequences appear for the first time.

7. Line270-277: Shouldn't the conclusion of this paragraph combine Table 1 and Figure 2? Figure 1 is only the genome construction process, not the conclusion of the paragraph.

8. Line295: “Pan-genomic Studies of Beef Cattle”, the introduction of the pan-genome lacks the necessary logic, so it is proposed to add an introduction to the assembly of the pan-genome in Part 3. In this section, it is recommended to introduce the relevant studies according to the pan-genome development history--iterative assembly, de novo assembly and graphic pan-genome assembly.

9. Line350: This part lacks the necessary connection with the genome and the pan-genome. It is recommended to describe the use of assembled genomes and pan-genomes to discover new functional genes or sequences.

10. The full paper mainly deals with genome assembly, pan-genome and identification of functional genes, and rarely deals with the introduction of genetic breeding. It is suggested to change the title of the article to suit the full paper.

Comments on the Quality of English Language

Moderate editing

Author Response

Dear Reviewer:

Thank you for your feedback and suggestions regarding our manuscript titled "Progress of the Beef Cattle Genome Project: Advances in Genome Assembly, Functional Gene Discovery, and Genetic Breeding." We appreciate your thoughtful evaluation and acknowledgment of the importance of this topic. We will carefully consider your comments as we undertake the major revisions required for the manuscript.

We have addressed your major concerns as follows:

The reviewer’s comment: 1. Line 105:the title is “Genome Assembly Technologies”, but line 132 is titled “Third-Generation Sequencing Assembly Software”,so is the third part mainly about assembling software or technology?

Response: We appreciate your attention to detail and your efforts to help us improve the clarity and structure of our work. First, we had to ensure that the “Genome Assembly Technologies” section comprehensively covered advances in assembly technologies, while the “Third Generation Sequencing Assembly Software” section focused specifically on the tools and software used for assembly in the context of third generation sequencing. Therefore, they needed to be discussed in one section.

The reviewer’s comment: 2. Line184-185:“four types of sequencing data required for T2T assembly” are mentioned in the paper. As we all know, the T2T genome assembly mainly relies on HIC, ONT ultra-long and HIFI reads sequencing data, why did you write about four data types with no references?

Response: We feel sorry for our carelessness. This section had references when it was written, but they were carelessly deleted during later revisions. The references have been reinserted into the text. (Please see lines 177 and 179 of the revised manuscript)

  1. Chaisson, M. J. P.; Huddleston, J.; Dennis, M. Y.; Sudmant, P. H.; Malig, M.; Hormozdiari, F.; Antonacci, F.; Surti, U.; Sandstrom, R.; Boitano, M.; Landolin, J. M.; Stamatoyannopoulos, J. A.; Hunkapiller, M. W.; Korlach, J.; Eichler, E. E. Resolving the complexity of the human genome using single-molecule sequencing. Nature 2015, 517, (7536), 608-611. https://doi.org/10.1038/nature13907
  2. Rautiainen, M.; Nurk, S.; Walenz, B. P.; Logsdon, G. A.; Porubsky, D.; Rhie, A.; Eichler, E. E.; Phillippy, A. M.; Koren, S. Telomere-to-telomere assembly of diploid chromosomes with Verkko. Nat. Biotechnol. 2023, 41, (10), 1474-1482. https://doi.org/10.1038/s41587-023-01662-6
  3. Cheng, H.; Concepcion, G. T.; Feng, X.; Zhang, H.; Li, H. Haplotype-resolved de novo assembly using phased assembly graphs with hifiasm. Nat. Methods 2021, 18, (2), 170-175. https://doi.org/10.1038/s41592-020-01056-5

The reviewer’s comment: 3. Line 192-193: please add reference documents to make your argument more convincing.

Response: Thank you for pointing this out. The contents of L182-187 are summarized in the article "Genome assembly in the telomere-to-telomere era" in Nature Reviews Genetics, which has been inserted in the L187 [30]. We feel sorry for our carelessness.

The reviewer’s comment: 4. Line204-209: The quality of genome assembly was assessed mainly for continuity, completeness, and accuracy, but for T2T genomes, the identification of centromere and telomere should also be added.

Response: Thank you for your valuable feedback on our manuscript. We appreciate your suggestion to include the identification of centromeres and telomeres as part of the quality assessment for T2T genomes. We added the relevant content and reference in lines 199-201.

  1. Lin, Y.; Ye, C.; Li, X.; Chen, Q.; Wu, Y.; Zhang, F.; Pan, R.; Zhang, S.; Chen, S.; Wang, X.; Cao, S.; Wang, Y.; Yue, Y.; Liu, Y.; Yue, J. quarTeT: a telomere-to-telomere toolkit for gap-free genome assembly and centromeric repeat identification. Hortic. Res. 2023, 10, (8). https://doi.org/10.1093/hr/uhad127

The reviewer’s comment: 5. Part 3 suggests adding an introduction to the Genome Annotation section.

Response: Thank you for your constructive feedback and valuable suggestions. We have added an introduction to the Genome Annotation section in Line 225-241 to provide a clearer context and enhance the flow of the manuscript.

The reviewer’s comment: 6. Line240: The full name should be written when EST sequences appear for the first time.

Response: Thank you for your insightful feedback on our manuscript. We appreciate your suggestion to include the full name for EST when it first appears. We have revised the text accordingly to state lines 250 "expressed sequence tag" upon its initial mention.

The reviewer’s comment: 7. Line270-277: Shouldn't the conclusion of this paragraph combine Table 1 and Figure 2? Figure 1 is only the genome construction process, not the conclusion of the paragraph.

Response: Thank you for your valuable feedback on our manuscript. We appreciate your careful review and for pointing out the oversight regarding the reference to the correct figure. We have revised the text to correctly reference Line 258 Figure 2 instead of Figure 1.

The reviewer’s comment: 8. Line295: “Pan-genomic Studies of Beef Cattle”, the introduction of the pan-genome lacks the necessary logic, so it is proposed to add an introduction to the assembly of the pan-genome in Part 3. In this section, it is recommended to introduce the relevant studies according to the pan-genome development history--iterative assembly, de novo assembly and graphic pan-genome assembly.

Response: Thank you very much for your valuable comments on our paper. We understand your suggestion to include an introduction to the assembly of the pan-genome in this section and to introduce relevant studies according to the development history of iterative assembly, de novo assembly, and graphical pan-genome assembly. However, our intention in writing this section was to focus primarily on the cutting-edge applications of the pan-genome in beef cattle. The studies we referenced are based on the latest graphical pan-genome assemblies. We are concerned that adding more historical background and detailed assembly processes as suggested might make this section overly lengthy. Therefore, we would like to keep the focus of this section on the advanced applications and recent research progress to highlight the practical value of pan-genomics in beef cattle genomic studies.

The reviewer’s comment: 9. Line350: This part lacks the necessary connection with the genome and the pan-genome. It is recommended to describe the use of assembled genomes and pan-genomes to discover new functional genes or sequences.

Response: Thank you for bringing up this issue. We have mentioned the use of assembled genomes and pan-genomes to discover new functional genes or sequences in Line 284-296 and 309-333. However, there is indeed limited research in this area, so we did not elaborate further in the subsequent sections. We hope this answer satisfies your query.

The reviewer’s comment: 10. The full paper mainly deals with genome assembly, pan-genome and identification of functional genes, and rarely deals with the introduction of genetic breeding. It is suggested to change the title of the article to suit the full paper.

Response: Thank you for raising this important question. After careful consideration, we have decided to change the title from "Beef Cattle Genome Project: Advances in Genome Assembly, Functional Genes Discovery and Genetic Breeding" to " Beef Cattle Genome Project: Advances in Genome Sequencing, Assembly and Functional Genes Discovery" This is because the content in the article, such as Genome Sequencing, Assembly and Functional Genes Discovery in beef cattle, indirectly impact genetic breeding practices rather than primarily focusing on breeding work in beef cattle.

Yours sincerely,

Dr. Weidong Deng

Yunnan Provincial Key Laboratory of Animal Nutrition and Feed, Faculty of Animal Science and Technology, Yunnan Agricultural University, Kunming 650201, China

Tel.: +86-871-65220375

Email: dengwd@ynau.edu.cn

Reviewer 3 Report

Comments and Suggestions for Authors

I have read this manuscript entitled "Beef Cattle Genome Project: Advances in Genome Assembly, Functional Genes Discovery and Genetic Breeding" from IJMS with No 3065515.

Based on the content provided, I believe this manuscript is very valuable, covering multiple aspects of beef cattle genome research, while also looking ahead to future trends and challenges. The article provides rich information and examples when describing various topics, but may need more detailed explanations or examples in certain areas to enhance readability and persuasiveness. Additionally, it is recommended to maintain a consistent style and tone throughout the article to ensure cohesiveness and fluency. Overall, this article offers valuable insights and perspectives, worthy of publication, but may require further minor modifications and refinements in certain sections as follows.

Major comments:

1.      Abstract: The abstract section is suggested to be polished to enhance logical connections.

  1. Introduction section: When introducing the Beef Cattle Genome Project, detailed descriptions of the specific goals and expected outcomes of the project are recommended. For example, specify which traits are the focus of the research and their impact on cattle breeding and agricultural production; reorganize paragraphs to make each topic clearer. For instance, separate the sections on technological developments and project objectives to help readers better understand the content of each part. Add transitional sentences to make logical connections between paragraphs more natural and smooth.
  2. Sequencing Technologies: Rearrange paragraphs to make the introduction of each sequencing technology clearer and more coherent. Consider describing the introduction of the technology and practical application cases separately to enhance logical flow.
  3. Genome Assembly Technologies: When emphasizing the importance of beef cattle genome assembly, further explanations on its practical applications and potential impacts in agriculture and breeding are suggested. Rearrange paragraphs to make each topic clearer. For example, describe different assembly strategies, software, and quality assessment separately to enhance logic and readability. Add transitional sentences to make logical connections between paragraphs more natural and smooth.
  4. Genomics Research Achievements in Beef Cattle: Ensure the timeliness and accuracy of cited references to reflect the latest research advancements. Add transitional sentences to make logical connections between paragraphs more natural.
  5. Functional Genes for Important Traits in Beef Cattle: Ensure smooth transitions between different sections and subsections. Sometimes the text jumps too abruptly, lacking clear connections. Introduce each subsection with a brief overview at the beginning, emphasizing the importance of each trait and its impact on beef cattle breeding. Some sentence structures are complex and can be simplified for easier understanding. Avoid overly long sentences by breaking them into shorter, more manageable parts. Clearly define technical terms when they first appear in the text to make it understandable to a broader audience.
  6. Conclusions and Perspective: When discussing future trends, mention the construction of a pan-genome for species and the integration of genomics with other technologies (such as single-cell sequencing and gene editing). These are areas worth paying attention to. Emphasize the importance of structural variations (SVs) in species adaptation studies and the application of multi-omics technologies in studying important economic traits. However, in the description of multi-omics studies, more emphasis should be placed on important and fundamental phenomics. Finally, the overall conclusion is somewhat lengthy and needs further condensation.

Minor comments:

  1. Figure 2 has low resolution, it is recommended to improve the pixel quality, especially changing the colors of the 3 lines at the top of Figure 2 to increase contrast and aesthetics.
  2. In L453-455: Due to longer sequence changes, SVs often experience stronger selection pressures than single-nucleotide variations, leading to greater disturbances in the genome [110-112]. Personally, I suggest adding 1-2 more references emphasizing the larger dosage effect of SV compared to SNPs. I only provide the following 2 references if you like.

Sudmant PH, Rausch T, Gardner EJ, et al. An integrated map of structural variation in 2,504 human genomes. Nature. 2015. 526(7571):75-81.

Yang N, Liu J, Gao Q, et al. Genome assembly of a tropical maize inbred line provides insights into structural variation and crop improvement. Nature Genetics. 2019. 51(6):1052-1059.

Author Response

Dear Reviewer:

Thank you very much for your thorough review and positive feedback on our manuscript entitled "Beef Cattle Genome Project: Advances in Genome Assembly, Functional Genes Discovery, and Genetic Breeding" (IJMS No. 3065515). We are pleased to hear that you find the manuscript valuable and appreciate your constructive comments. Additionally, we will ensure a consistent style and tone throughout the article to maintain cohesiveness and fluency. Your insights and suggestions are highly valuable to us, and we will make the necessary minor modifications and refinements to improve the manuscript further. Thank you again for your support and recommendations.

We have addressed your major and minor comments as follows:

Major comments:

The reviewer’s comment: 1. Abstract: The abstract section is suggested to be polished to enhance logical connections.

Response: Thanks for your suggestion. We have tried our best to polish the abstract section in the revised manuscript.

We have changed Lines 15-29 into: “Beef is a major global source of protein, playing an essential role in the human diet. The worldwide production and consumption of beef continue to rise, reflecting a significant trend. However, despite the critical importance of beef cattle resources in agriculture, the diversity of cattle breeds faces severe challenges, with many breeds at risk of extinction. The initiation of the Beef Cattle Genome Project is crucial. By constructing a high-precision functional annotation map of the genome, it becomes possible to analyze the genetic mechanisms underlying important traits in beef cattle, laying a solid foundation for breeding more efficient and productive cattle breeds. This review details advances in genome sequencing and assembly technologies, iterative upgrades of the beef cattle reference genome, and its application in pangenome research. Additionally, it summarizes relevant studies on the discovery of functional genes associated with key traits in beef cattle, such as growth, meat quality, reproduction, polled traits, disease resistance, and environmental adaptability. Finally, the review explores the potential of telomere-to-telomere (T2T) genome assembly, genome structural variation (SV), and multi-omics techniques in future beef cattle genetic breeding. These advancements collectively offer promising avenues for enhancing beef cattle breeding and improving genetic traits.”

The reviewer’s comment: 2. Introduction section: When introducing the Beef Cattle Genome Project, detailed descriptions of the specific goals and expected outcomes of the project are recommended. For example, specify which traits are the focus of the research and their impact on cattle breeding and agricultural production; reorganize paragraphs to make each topic clearer. For instance, separate the sections on technological developments and project objectives to help readers better understand the content of each part. Add transitional sentences to make logical connections between paragraphs more natural and smooth.

Response: Thank you for your detailed and insightful feedback on our manuscript. we polished the language in the Introduction section Line 33-68.

The reviewer’s comment: 3. Sequencing Technologies: Rearrange paragraphs to make the introduction of each sequencing technology clearer and more coherent. Consider describing the introduction of the technology and practical application cases separately to enhance logical flow.

Response: Thank you for your valuable feedback on the Sequencing Technologies section of our manuscript. We appreciate your suggestion, but this section describes the first, second and third generation sequencing in three paragraphs, so I modified some of the content to make the paragraph logic clearer.

We have changed Line 83-84 “The early 21st century, Next Generation DNA Sequencing (NGS) revolutionized genomics by significantly reducing costs and increasing throughput.” into Line 80-81 “In the early 21st century, Next Generation DNA Sequencing (NGS) revolutionized genomics by significantly reducing costs and increasing throughput.”

The reviewer’s comment: 4. Genome Assembly Technologies: When emphasizing the importance of beef cattle genome assembly, further explanations on its practical applications and potential impacts in agriculture and breeding are suggested. Rearrange paragraphs to make each topic clearer. For example, describe different assembly strategies, software, and quality assessment separately to enhance logic and readability. Add transitional sentences to make logical connections between paragraphs more natural and smooth.

Response: We appreciate the valuable feedback from the reviewer on the Genome Assembly Technologies section of our manuscript. Based on your suggestions, we incorporate practical applications and potential impacts, as well as enhance the logical flow with transitional sentences (Please see lines 101-241 of the revised manuscript).

The reviewer’s comment: 5. Genomics Research Achievements in Beef Cattle: Ensure the timeliness and accuracy of cited references to reflect the latest research advancements. Add transitional sentences to make logical connections between paragraphs more natural.

Response: Thank you for your valuable feedback on the manuscript. We have revised section 4 to ensure the timeliness and accuracy of cited references, reflecting the latest advancements in beef cattle genomics. Transitional sentences have been added to enhance the logical flow between paragraphs, improving readability and coherence (Please see lines 242-340 of the revised manuscript).

The reviewer’s comment: 6. Functional Genes for Important Traits in Beef Cattle: Ensure smooth transitions between different sections and subsections. Sometimes the text jumps too abruptly, lacking clear connections. Introduce each subsection with a brief overview at the beginning, emphasizing the importance of each trait and its impact on beef cattle breeding. Some sentence structures are complex and can be simplified for easier understanding. Avoid overly long sentences by breaking them into shorter, more manageable parts. Clearly define technical terms when they first appear in the text to make it understandable to a broader audience.

Response: We appreciate your thorough review and suggestions for improving the clarity and coherence of the text. We acknowledge the need for smoother transitions between different sections and subsections. To address this, we will introduce each subsection with a brief overview highlighting the significance of each trait in beef cattle breeding. This approach will ensure a clearer connection between the subsections, facilitating a more coherent flow of information throughout the manuscript. Regarding the complexity of sentence structures, we understand the importance of clarity and accessibility for a broader audience. We will revise and simplify complex sentences by breaking them into shorter, more digestible parts (Please see lines 356-360, 377-378, 391-392 of the revised manuscript).

The reviewer’s comment: 7. Conclusions and Perspective: When discussing future trends, mention the construction of a pan-genome for species and the integration of genomics with other technologies (such as single-cell sequencing and gene editing). These are areas worth paying attention to. Emphasize the importance of structural variations (SVs) in species adaptation studies and the application of multi-omics technologies in studying important economic traits. However, in the description of multi-omics studies, more emphasis should be placed on important and fundamental phenomics. Finally, the overall conclusion is somewhat lengthy and needs further condensation.

Response: We appreciate your feedback and have condensed the content to make it more concise and focused. We have changed Line 425-472 into Line 421-450, and removed the number of references concurrently.

Minor comments:

The reviewer’s comment: 1. Figure 2 has low resolution, it is recommended to improve the pixel quality, especially changing the colors of the 3 lines at the top of Figure 2 to increase contrast and aesthetics.

Response: We appreciate your suggestion to improve the pixel quality of Figure 2, particularly enhancing the resolution and adjusting the colors of the three lines at the top to increase contrast and improve aesthetics. We will carefully address these aspects in the revised version of the manuscript to ensure clarity and visual appeal.

The reviewer’s comment: 2. In L453-455: Due to longer sequence changes, SVs often experience stronger selection pressures than single-nucleotide variations, leading to greater disturbances in the genome [110-112]. Personally, I suggest adding 1-2 more references emphasizing the larger dosage effect of SV compared to SNPs. I only provide the following 2 references if you like.

Sudmant PH, Rausch T, Gardner EJ, et al. An integrated map of structural variation in 2,504 human genomes. Nature. 2015. 526(7571):75-81.

Yang N, Liu J, Gao Q, et al. Genome assembly of a tropical maize inbred line provides insights into structural variation and crop improvement. Nature Genetics. 2019. 51(6):1052-1059.

Response: We appreciate your feedback and have condensed the content to make it more concise and focused. After careful consideration, we have accepted the suggestions from other reviewers to reduce the content of The Conclusions and Perspective section and remove the references. This section will now focus on being concise.

Yours sincerely,

Dr. Weidong Deng

Yunnan Provincial Key Laboratory of Animal Nutrition and Feed, Faculty of Animal Science and Technology, Yunnan Agricultural University, Kunming 650201, China

Tel.: +86-871-65220375

Email: dengwd@ynau.edu.cn

Reviewer 4 Report

Comments and Suggestions for Authors

The review deals with the cattle genome research. The topic is of interest, review summarizes recent knowledge in the field. Minor revisions are needed.

Try to shorten the title.

Some sentences are too long and difficult to understand, rr. 57-61; 251-257, e.g.

Section 4.1, give all countries in the consortium, or none.

Throughout the text, the abbreviations of genes in italic.

Tab. 1, the title announces beef cattle, but there are also dual Original Braunvieh and milk Holstein breeds. Correct the title.

R. 278 and 283, use the same name of the breed.

Main comment: mention concisely, 1-2 paragraphs, the perspective and significance of whole genome sequencing in evaluation of breeding value and selection.

Formal errors:

In the list of authors, the last name is may missing.

Author Response

Dear Reviewer:

Thank you for your thoughtful review and valuable feedback. We appreciate your positive comments regarding the topic's relevance and the summary of recent advancements in cattle genome research. We will carefully address the minor revisions you've suggested to ensure the clarity and accuracy of the manuscript. Your insights will undoubtedly contribute to improving the quality and impact of our review. Thank you once again for your time and constructive input.

We have addressed your major and minor concerns as follows:

The reviewer’s comment: Try to shorten the title.

Response: Thank you for raising this important question. After careful consideration, we have decided to change the title from "Beef Cattle Genome Project: Advances in Genome Assembly, Functional Genes Discovery and Genetic Breeding" to " Beef Cattle Genome Project: Advances in Genome Sequencing, Assembly and Functional Genes Discovery" This is because the content in the article, such as Genome Sequencing, Assembly and Functional Genes Discovery in beef cattle, indirectly impact genetic breeding practices rather than primarily focusing on breeding work in beef cattle.

The reviewer’s comment: Some sentences are too long and difficult to understand, rr. 57-61; 251-257, e.g.

Response: Thank you for your suggestion. We have revised the sentence as per your recommendation to improve clarity and readability. Your feedback is greatly appreciated (Please see lines 55-58, 244-246 of the revised manuscript).

The reviewer’s comment: Section 4.1, give all countries in the consortium, or none.

Response: Thank you for your feedback. We have revised the manuscript accordingly, as indicated in line 244. The list of countries involved has been omitted to streamline the text. Please let us know if there are any further adjustments needed. Your comments have been valuable in improving the clarity of our manuscript.

The reviewer’s comment: Throughout the text, the abbreviations of genes in italic.

Response: Throughout the text, we have corrected the formatting of gene abbreviations to italicize them as per your suggestion. This will enhance clarity and conformity in gene notation.

The reviewer’s comment: Tab. 1, the title announces beef cattle, but there are also dual Original Braunvieh and milk Holstein breeds. Correct the title.

Response: Regarding Table 1, we apologize for the oversight in the title. As the focus of the article is primarily on beef cattle, we have removed references to Braunvieh and milk Holstein breeds from the title to accurately reflect the content.

The reviewer’s comment: R. 278 and 283, use the same name of the breed.

Response: We appreciate your observation regarding the consistency in breed names. We have revised line 288 to replace "Chinese yellow cattle" with "Chinese taurine cattle" to maintain clarity and accuracy throughout the manuscript.

The reviewer’s comment: Main comment: mention concisely, 1-2 paragraphs, the perspective and significance of whole genome sequencing in evaluation of breeding value and selection.

Response: Thank you for your valuable feedback. Based on your suggestion, we have added a concise section discussing the perspective and significance of whole genome sequencing in the evaluation of breeding value and selection on Line 345-348.

The reviewer’s comment: Formal errors: In the list of authors, the last name is may missing.

Response: Concerning formal errors in the list of authors, we have reviewed and ensured that all authors' last names are correctly listed without any omissions. So we have deleted 'and' in Line 5.

Yours sincerely,

Dr. Weidong Deng

Yunnan Provincial Key Laboratory of Animal Nutrition and Feed, Faculty of Animal Science and Technology, Yunnan Agricultural University, Kunming 650201, China

Tel.: +86-871-65220375

Email: dengwd@ynau.edu.cn

Round 2

Reviewer 1 Report

Comments and Suggestions for Authors

Thanks for author’s responses. However, the format of some references is not suitable for this Journal, and abbreviated journal name should be used in Reference section.

Comments on the Quality of English Language

Moderate editing of English language required.

Reviewer 2 Report

Comments and Suggestions for Authors

Author responded all my issues.

Reviewer 3 Report

Comments and Suggestions for Authors

Now I recommend accept at the current form.
